# Foggy perception slows us down

**Paolo Pretto[1]\*[†], Jean-Pierre Bresciani[2],[3][†], Gregor Rainer[3], Heinrich H Bülthoff[1]\***

[1]Department of Human Perception, Cognition and Action, Max Planck Institute for Biological Cybernetics, Tübingen, Germany; [2]Psychology and NeuroCognition Laboratory, University Pierre Mendès-France and CNRS, Grenoble, France; [3]Department of Medicine, University of Fribourg, Fribourg, Switzerland

**Abstract** Visual speed is believed to be underestimated at low contrast, which has been proposed as an explanation of excessive driving speed in fog. Combining psychophysics measurements and driving simulation, we confirm that speed is underestimated when contrast is reduced uniformly for all objects of the visual scene independently of their distance from the viewer. However, we show that when contrast is reduced more for distant objects, as is the case in real fog, visual speed is actually overestimated, prompting drivers to decelerate. Using an artificial anti-fog—that is, fog characterized by better visibility for distant than for close objects, we demonstrate for the first time that perceived speed depends on the spatial distribution of contrast over the visual scene rather than the global level of contrast per se. Our results cast new light on how reduced visibility conditions affect perceived speed, providing important insight into the human visual system.

**\*For correspondence:** paolo. pretto@tuebingen.mpg.de (PP); heinrich.buelthoff@tuebingen. mpg.de (HHB)

[†]These authors contributed equally to this work

**Competing interests:** The authors have declared that no competing interests exist

**Reviewing editor**: Jody C Culham, University of Western Ontario, Canada

## Introduction

Visual contrast is usually referred to as the difference in brightness between an object and the background (*Hofstetter et al., 2000*). Classical vision research experiments systematically investigated how visual contrast affects objects motion perception (*Thompson, 1982*; *Stone and Thompson, 1992*; *Blakemore and Snowden, 1999*; *Anstis, 2003*). These studies have shown that the perceived speed of two-dimensional moving objects—for example, plaid patterns on a computer screen—is underestimated when visual contrast is reduced. More recent studies based on driving scenarios have suggested that the underestimation of visual speed at low contrast applies also to perceived self-motion in three-dimensional environments (*Snowden et al., 1998*; *Horswill and Plooy, 2008*; *Owens et al., 2010*). This finding was proposed—and is still considered—as a possible explanation for excessive driving speed in fog.

In the abovementioned studies, and more generally in all studies having assessed the effect of visual contrast on motion perception, contrast was reduced uniformly for all objects of the visual scene, irrespective of their distance from the observer. While uniform contrast reduction is a valid model to assess the perception of two-dimensional patterns moving on a computer screen, it is a poor model to investigate how an atmospheric phenomenon like fog affects motion perception in three-dimensional environments. Specifically, fog alters visual contrast because tiny water droplets suspended in the air are interposed between the observer and the surrounding objects. The quantity of droplets increases along the line-of-sight as distance from the observer increases. As a consequence, the contrast of the visual scene decreases with distance, and visibility is better for close than for distant objects. Therefore, the distance-dependent attenuation of contrast experienced in fog is obviously very different from the uniform reduction traditionally adopted in vision research. Uniform contrast reduction resembles more closely what one could experience when looking at the environment through a dirty glass or a foggy windshield.

To date, the effects of distance-dependent contrast reduction on motion perception are unknown. In other words, we still do not know how fog affects perceived self-motion. Here, we tested the

**eLife digest** The ways people respond to conditions of reduced visibility is a central topic in vision research. Notably, it has been shown that people tend to underestimate speeds when visibility is reduced equally at all distances, as for example, when driving with a fogged up windshield. But what happens when the visibility decreases as you look further into the distance, as happens when driving in fog? Fortunately, as new research reveals, people tend to overestimate their speed when driving in fog-like conditions, and show a natural tendency to drive at a slower pace.

Pretto et al. performed a series of experiments involving experienced drivers and high-quality virtual reality simulations. In one experiment, drivers were presented with two driving scenes and asked to guess which scene was moving faster. In the reference scene, the car was driving at a fixed speed through a landscape under conditions of clear visibility; in the test scene, it was moving through the same landscape, again at a fixed speed, but with the visibility reduced in different ways. The experiments showed that drivers overestimated speeds in fog-like conditions, and they underestimated speeds when the reduction in visibility did not depend on distance. Further experiments confirmed that these perceptions had an influence on driving behaviour: drivers recorded an average speed of 85.1 km/hr when the visibility was good, and this dropped to 70.9 km/hr in severe fog. However, when visibility was reduced equally at all distances, as happens with a fogged up windshield, the average driving speed increased to 101.3 km/hr.

Based on previous work, Pretto et al. developed the theory that the perception of speed is influenced by the relative speeds of the visible regions in the scene. When looking directly into the fog, visibility is strongly reduced in the distant regions, where the relative motion is slow, and is preserved in the near regions, where the motion is fast. This visibility gradient would lead to speed overestimation. To test this theory, the experiments were repeated with new drivers under three different conditions: good visibility, fog, and an artificial situation called 'anti-fog' in which visibility is poor in the near regions and improves as the driver looks further into the distance. As predicted, the estimated speed was lower in anti-fog than in clear visibility and fog. Conversely, the driving speed was 104.4 km/hr in anti-fog compared with 67.9 km/hr in good visibility and 51.3 km/hr in fog.

Overall, the results show that the perception of speed is influenced by spatial variations in visibility, and they strongly suggest that this is due to the relative speed contrast between the visible and covert areas within the scene.

perceptual and behavioural effects of distance-dependent contrast reduction—that is, fog—on speed perception. We compared these effects with those of the distance-independent—that is, uniform—contrast reduction that has been used in previous studies to simulate fog (*Snowden et al., 1998*; *Horswill and Plooy, 2008*; *Owens et al., 2010*). To perform these experiments, we used a state-of-the-art virtual reality setup allowing us to realistically simulate fog in an ecological driving scenario (*Loomis et al., 1999*).

## Results

In the first experiment, we used a standard psychophysical procedure to test how contrast affects perceived visual speed. Twelve experienced drivers were presented with pairs of driving scenes and instructed to estimate which scene moved faster (*Figure 1A*). One of the scenes (reference) had clear visibility and moved at one of three target speeds (40, 60, or 90 km/hr). The other scene (test) had either clear or reduced visibility, and its speed was adjusted for each trial using a Bayesian adaptive method (*Kontsevich and Tyler, 1999*). This method allowed us to determine the point of subjective equality (PSE) as well as the just-noticeable difference (JND). The PSE corresponded to the speed at which the two scenes were perceived as moving equally fast. Therefore, PSEs higher than the actual speed of the reference scene indicated speed underestimation, whereas PSEs lower than the speed of the reference scene indicated speed overestimation. The JND corresponded to the smallest detectable difference between two different speeds. High JNDs indicated low discrimination sensitivity, whereas low JNDs indicated high discrimination sensitivity.

The contrast of the scene was reduced either in a distance-dependent manner, as would happen in natural fog, or in a distance-independent manner, as has been done in previous experiments.

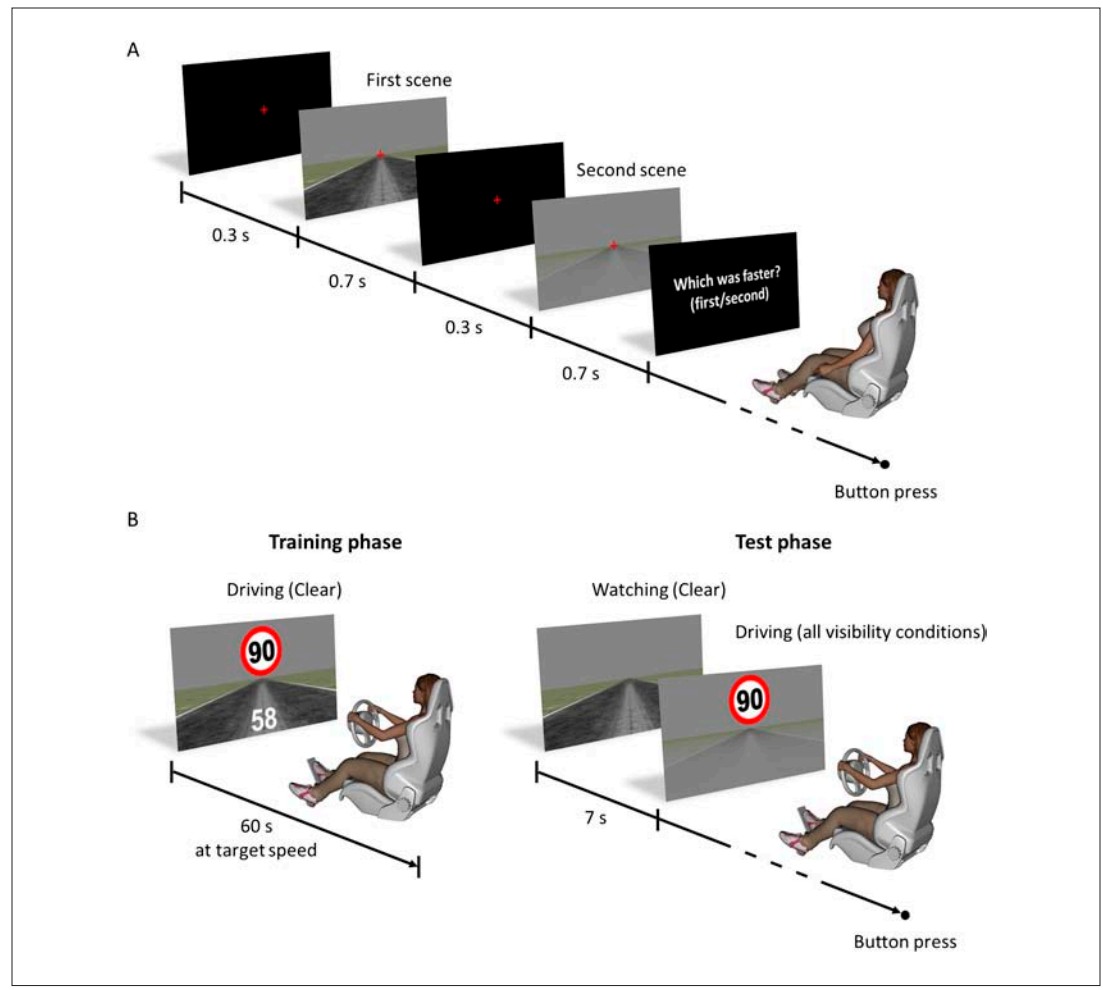

**Figure 1**. Experimental design and time course of trials. (**A**) Experiments 1 and 3: for each trial, the first scene was presented for 700 ms, which included a 100-ms fade-in phase at the beginning and a 100-ms fade-out phase at the end. The second scene was presented 300 ms after the end of the first scene and had the same temporal structure as the first one. Participants had to fixate a central cross for the whole duration of the trial. The order of presentation of the reference and test scene was randomized. (**B**) Experiments 2 and 4: three driving sessions (i.e., one per target speed) were performed in random order for experiment 2, and one session for experiment 4. Before each test session, the drivers performed a training phase in which a numerical feedback indicated the driving speed when it did not match the target speed (white digits at the bottom of the screen, left panel). In each training phase, the drivers had to drive a total of 5 min at target speed. In addition, at the beginning of each test trial, the scene was shown for 7 s moving at target speed with clear visibility (memory refresher).

The difference between these two types of contrast reduction is represented in *Figure 2B and C*. Moderate and severe levels of reduction were implemented for each type of contrast alteration. Importantly, for each level of reduction, the overall visual contrast was the same for distance-dependent and distance-independent alteration (Root mean square [RMS] contrast = 0.31 for moderate visibility reduction and 0.19 for severe visibility reduction, vs 0.46 for clear visibility). In total, the experiment consisted of five visibility conditions: clear (no contrast reduction), moderate and severe fog (distance-dependent contrast reduction), moderate and severe uniform reduction (distance-independent contrast reduction).

Reducing the contrast of the visual scene altered speed perception [$F_{(4,44)}$ = 52.086, p<0.001, $\eta_G^2$ = 0.61]. However, as shown in *Figure 3A*, perceived speed was affected differently by the two types of contrast reduction. Specifically, when contrast reduction depended on distance, participants matched lower speeds (PSE mean = 54.7 and 41.7 km/hr for moderate and severe fog, respectively) to the

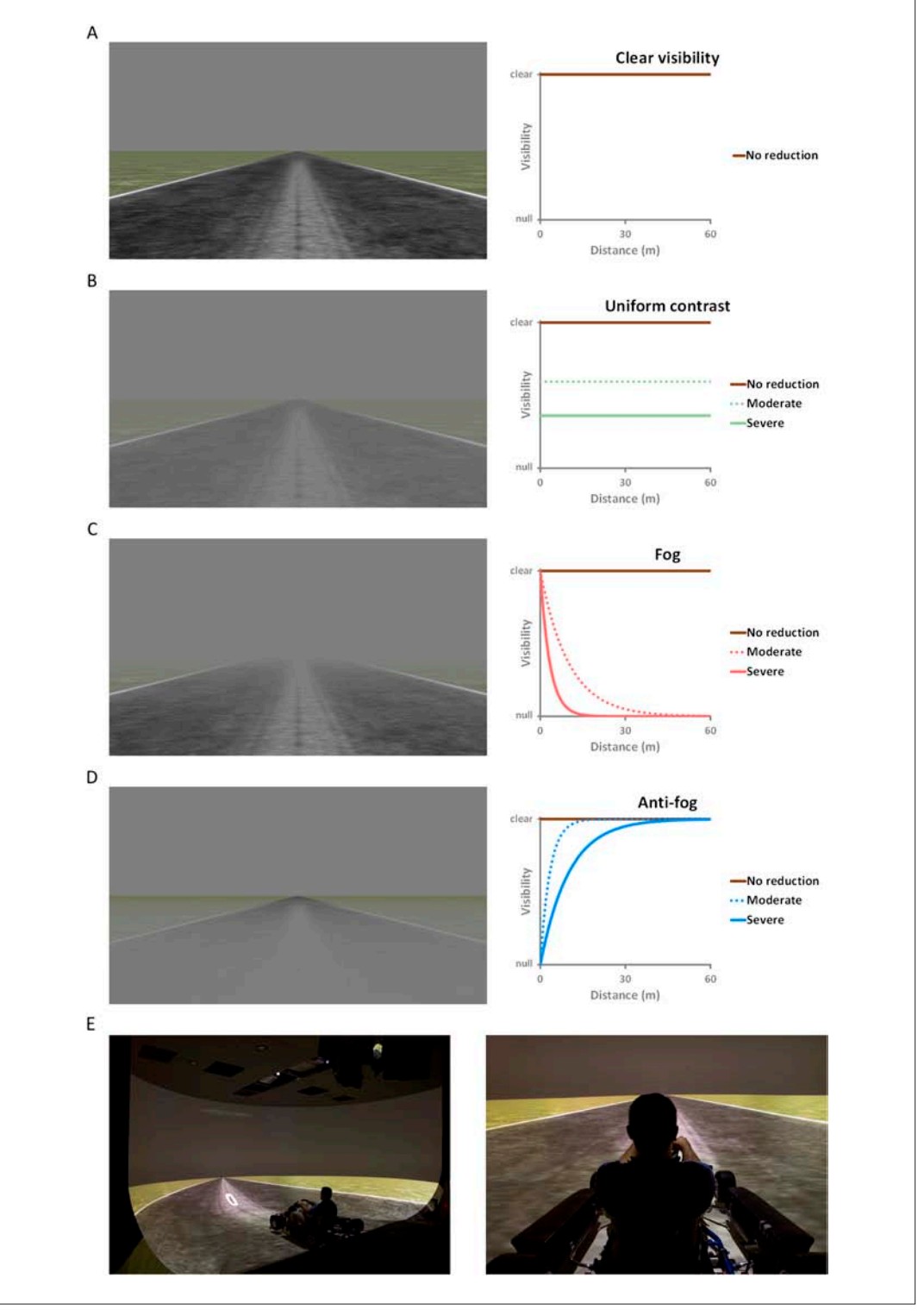

**Figure 2**. Visibility conditions. (**A**) Clear weather conditions (clear visibility): contrast is unaltered and the visibility is optimal in all directions (brown line). (**B**) Distance-independent contrast reduction (uniform contrast): visibility drops equally for all objects of the visual scene, irrespective of their distance from the observer (green lines). (**C**) Distance-dependent contrast reduction (fog): visibility is good for close objects, and worsens as distance from the observer increases (red lines). (**D**) Reversed distance-dependent contrast reduction (anti-fog): visibility is poor for close objects, and improves as distance from the observer increases (blue lines). (**E**) Pictures of the actual setup: side view (left) and driver's view (right).

perceived speed under clear visibility (mean = 65 km/hr). This indicates that natural fog led to an over-estimation of speed (equivalent to 76.4 and 94.3 km/hr for moderate and severe fog, respectively; *Figure 3A*). Conversely, when visibility was reduced in a distance-independent manner, higher speeds (PSE mean = 95.5 and 95.2 km/hr for moderate and severe uniform reduction, respectively) were matched to the perceived speed under clear visibility. Therefore, speed was underestimated with uniform contrast reduction (perceived speed equivalent to 41.4 and 41.5 km/hr for moderate and severe reduction, respectively). Each condition differed from each of the others (p<0.05), except for uniform-moderate and uniform-severe reduction that did not differ significantly from one another. The same pattern of results was observed for all three target speeds. These results show that the two types of contrast reduction gave rise to opposite perceptual effects.

Reducing visibility also affected speed discrimination sensitivity [$F_{(4,44)}$ = 29.58, p<0.001, $\eta_G^2$ = 0.37], which was significantly lower (higher JNDs) in the two conditions in which visibility was reduced in a distance-independent manner (mean = 0.58 km/hr for both moderate and severe uniform reduction), as compared to the other three conditions that did not differ from one another (mean = 0.41, 0.40, and 0.42 km/hr for clear, moderate, and severe fog, respectively). This indicates that the participants had more difficulties estimating the driving speed when visibility reduction was independent of distance. This reduction of speed discrimination sensitivity might result from the fact that distance-independent contrast alteration is seldom encountered in real life. Specifically, although entering a car with a foggy windshield is common, especially due to cold weather, drivers usually demist their windshield before starting to drive. Unfortunately, drivers' options to reduce environmental fog are much more restricted.

In the second experiment, we tested whether the perceptual changes measured in the first experiment affect actual driving behaviour. Ten experienced drivers who did not participate in the first experiment were instructed to drive at target speeds with either clear or reduced visibility. The visibility conditions and target speeds were the same as in experiment 1. In a preliminary training phase, drivers learned to reach and maintain each target speed with clear visibility (*Figure 1B*, left pane). In the test phase, drivers freely controlled their driving speed with the gas pedal, and pressed a button when thinking they were driving at target speed (*Figure 1B*, right pane). To refresh drivers' memory, the scene was shortly presented moving at the target speed with clear visibility at the beginning of each trial.

Driving speed was affected by the contrast of the visual scene [$F_{(4,36)}$ = 43.18, p<0.001, $\eta_G^2$ = 0.44]. However, as for perceived speed, the direction of the effect depended on the type of contrast reduction. As compared to clear visibility (mean = 85.1 km/hr), the participants drove significantly slower with distance-dependent contrast reduction (mean = 77.3 and 70.9 km/hr for moderate and severe fog, respectively), but faster with distance-independent reduction (mean = 93.6 and 101.3 km/hr for moderate and severe uniform reduction, respectively; see *Figure 3B*). The five visibility conditions differed significantly from one another (p<0.05), and the same pattern was observed for all three target speeds. These behavioural results are consistent with the perceptual results of the first experiment and can be interpreted as follows: distance-dependent contrast reduction induces an overestimation of visual speed that prompts drivers to drive slower, whereas distance-independent contrast reduction evokes an underestimation of visual speed, prompting drivers to drive faster.

The results of the first two experiments highlight that the distance-dependent contrast reduction experienced in fog evokes perceptual and behavioural effects that are radically opposite to those resulting from the distance-independent contrast reduction traditionally adopted in vision research. Such a radical difference is striking because in our experiments, the two types of contrast reduction resulted in the same global attenuation of visual contrast. This suggests that the global level of contrast is not the only factor affecting perceived speed. The spatial distribution of contrast over the visual scene also seems to play a critical role. Yet, what are the perceptual mechanisms underlying speed overestimation in fog?

In fog, visibility—that is, contrast—is reduced with distance. Therefore, when the driver looks straight ahead, the regions of better visibility in the proximity of the vehicle are typically viewed in peripheral vision, whereas the distant regions where visibility is impaired fall in the centre of the driver's visual field (*Snowden and Freeman, 2004*). In this situation, fog acts as a mask obscuring the central region of the visual field. Recently, we have shown that speed perception strongly depends on the visible portion of the visual field (*Pretto et al., 2009*). In particular, we found that speed is overestimated when the central area is occluded, and underestimated when it is the peripheral area that is masked. A similar phenomenon could explain why speed is overestimated in fog, that is, when

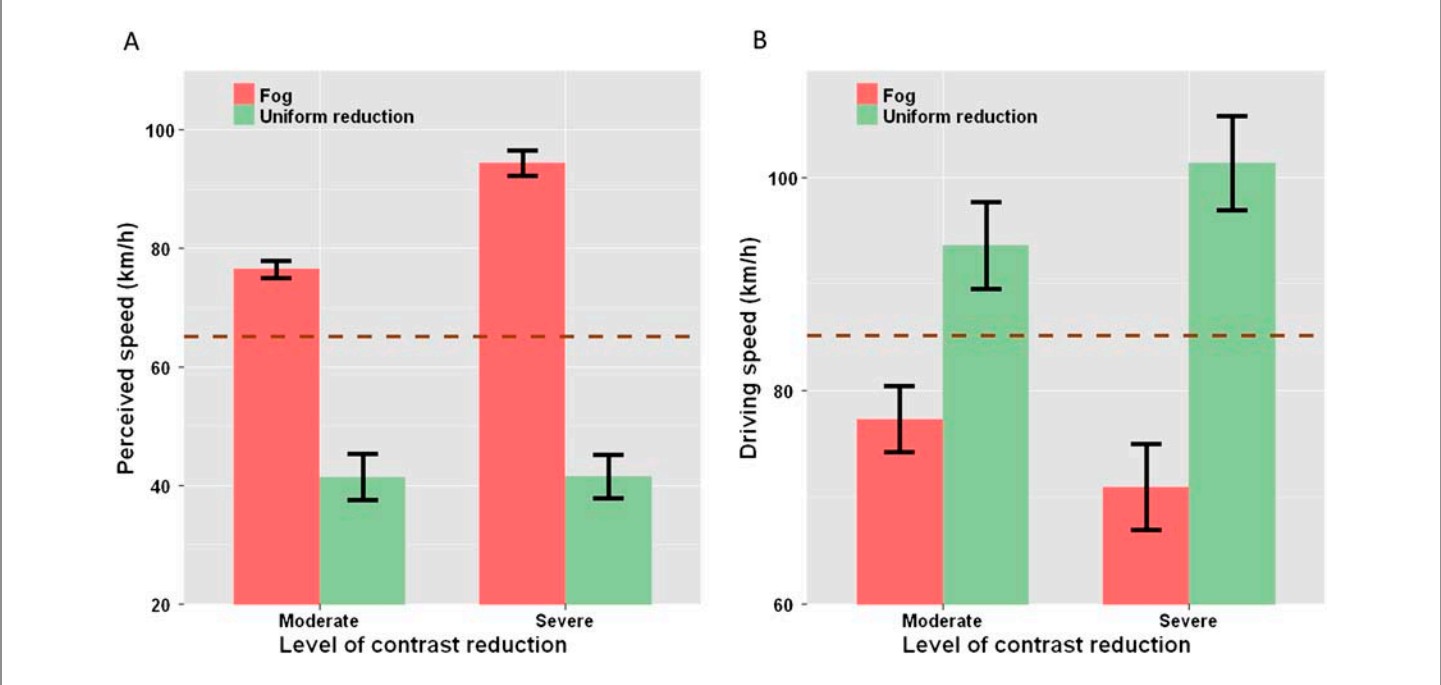

**Figure 3**. Opposite effects of distance-dependent and distance-independent contrast reduction. Experiments 1 and 2. (**A**) Mean perceived driving speed across subjects as a function of visibility: for each subject, PSE values were averaged across the three target speeds (i.e., 40, 60, and 90 km/hr), then perceived speed was calculated using the following equation: Speed$_{perceived}$ = PSE$_{clear}$ + PSE$_{clear}$ × ln(PSE$_{clear}$/PSE$_{reduced visibility}$). As compared to clear visibility (brown dashed line), speed was overestimated with distance-dependent visibility reduction (red bars) and underestimated with distance-independent visibility reduction (green bars). (**B**) Mean produced driving speed across subjects as a function of visibility: for each subject, measured speed values were averaged across the three target speeds. As compared to their driving speed with clear visibility (brown dashed line), drivers drove slower with distance-dependent visibility reduction (red bars) and faster with distance-independent visibility reduction (green bars). In both (**A**) and (**B**), the error bars represent the standard error of the mean. PSE: point of subjective equality.

contrast is reduced in a distance-dependent manner. More specifically, speed overestimation in fog could result from the relative contrast between the central and peripheral areas of the visual field.

To test this hypothesis, we created an anti-fog, that is, a distance-dependent contrast reduction characterized by an increase of visibility with distance (see *Figure 2D*). With anti-fog, visibility was better for distant than for close objects, that is, visibility was good for the portion of road situated at a distance ahead, and poor for the direct surroundings of the vehicle. In other words, normal fog and anti-fog resulted in opposite spatial distributions of contrast over the visual scene, although the global contrast reduction was the same for both fog types. We hypothesized that if the relative contrast between central and peripheral areas of the visual field underlies the speed overestimation observed with fog, then speed should be underestimated with the anti-fog.

In the third experiment, we used the same psychophysical procedure as in the first experiment to compare the effect of natural fog and anti-fog on perceived speed. Ten experienced drivers who had not participated in the first two experiments were presented with pairs of driving scenes and instructed to estimate which scene moved faster (see methods of experiment 1). Three visibility conditions were used: clear, fog, and anti-fog. Only one target speed was used (60 km/hr) as the first two experiments revealed the same pattern of results for all three target speeds.

As shown in *Figure 4A*, perceived speed depended on visibility [$F_{(2,18)}$ = 65.64, p<0.001, $\eta_G^2$ = 0.81]. With fog, lower speeds (mean PSE = 47.7 km/hr) were matched to the perceived speed with clear visibility (mean PSE = 60.1 km/hr), whereas higher speeds were matched to it with anti-fog (mean PSE = 121.6 km/hr). Therefore, as compared to clear visibility, the scenes were perceived as moving faster with fog (equivalent to 74.3 km/hr, *Figure 4A*) and slower with anti-fog (equivalent to 19 km/hr). All three visibility conditions differed significantly from one another (p<0.05). Visibility also affected speed discrimination sensitivity [$F_{(2,18)}$ = 82.85, p<0.001, $\eta_G^2$ = 0.79]. JND was twice as high in the anti-fog condition (mean = 0.543 km/hr) than in the other two conditions that did not differ from one another

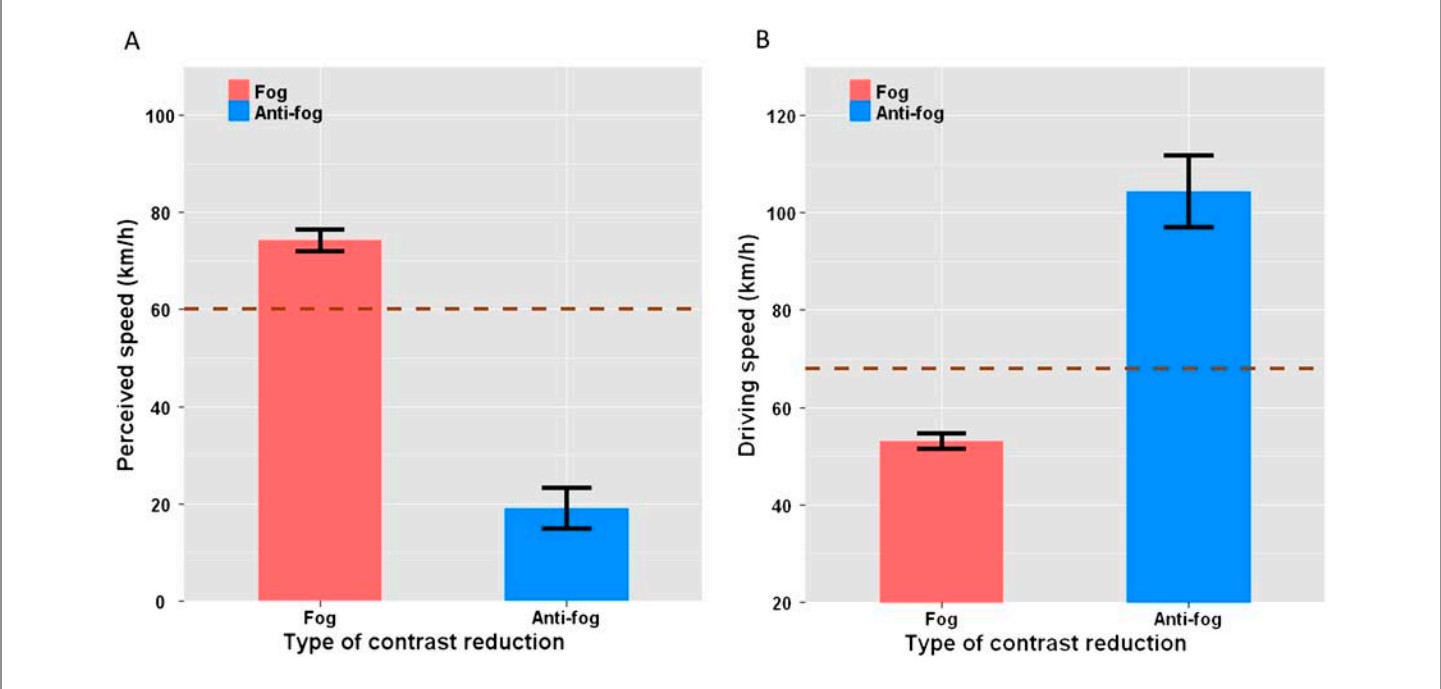

**Figure 4**. Opposite effects of fog and anti-fog. Experiments 3 and 4. (**A**) Mean perceived driving speed across subjects as a function of visibility: Perceived speed was calculated from the measured PSEs using the following equation: $\text{Speed}_{\text{perceived}} = \text{PSE}_{\text{clear}} + \text{PSE}_{\text{clear}} \times \ln(\text{PSE}_{\text{clear}}/\text{PSE}_{\text{reduced visibility}})$. As compared to clear visibility (brown dashed line), speed was overestimated when visibility was better for close than for distant objects, that is, in fog (red bars), and underestimated when visibility was better for distant than for close objects, that is, anti-fog (blue bars). (**B**) Mean produced driving speed across subjects as a function of visibility: As compared to their driving speed with clear visibility (brown dashed line), drivers drove slower when visibility was better for close than for distant objects, that is, in fog (red bars), and faster when visibility was better for distant than for close objects, that is, anti-fog (blue bars). In both (**A**) and (**B**), the error bars represent the standard error of the mean. PSE: point of subjective equality.

(mean = 0.248 and 0.225 km/hr for clear and fog, respectively). This indicates that estimating driving speed was more difficult in the anti-fog condition, which likely results from the completely artificial nature of this type of contrast reduction. Indeed, anti-fog is a type of contrast alteration that never occurs in real life. In that respect, it is interesting to mention that reduced speed discrimination performance was observed with both unusual types of contrast reduction, namely anti-fog here and uniform contrast reduction in the first experiment.

The same 10 participants took part in a fourth experiment where they were instructed to drive at target speeds with either clear or reduced visibility (same driving procedure as experiment 2). This experiment aimed to compare the effects of fog and anti-fog on produced driving speed. The visibility and speed conditions were identical to those used in experiment 3. The order of experiments 3 and 4 was counterbalanced between participants.

Driving speed was affected by visibility [$F_{(2,18)} = 39.99$, $p<0.001$, $\eta_G^2 = 0.71$]. Specifically, as compared to clear visibility (mean = 67.9 km/hr), participants drove slower with fog (mean = 53.1 km/hr) and faster with anti-fog (mean = 104.4 km/hr, see ***Figure 4B***). All three conditions differed from one another ($p<0.05$). Visibility also affected the variability of driving speed [$F_{(2,18)} = 9.56$, $p<0.01$, $\eta_G^2 = 0.33$], which was twice as large in the anti-fog condition (mean = 17.6 km/hr) as in the other two conditions that did not differ from one another (mean = 9 and 7.5 km/hr for clear visibility and fog, respectively).

## Discussion

Previous studies suggested that the speed of visual motion in depth is underestimated when the global level of contrast is reduced (***Snowden et al., 1998***; ***Horswill and Plooy, 2008***; ***Owens et al., 2010***). These studies were directly inspired by classical vision research experiments that assessed the effect of contrast on the perceived speed of two-dimensional objects on a monitor (***Thompson, 1982***; ***Stone***

*and Thompson, 1992*; *Blakemore and Snowden, 1999*; *Anstis, 2003*). In the present study, we reproduced this perceptual bias, showing that visual speed is indeed underestimated when contrast is reduced in a distance-independent manner. However, we show that this is only part of the whole picture. In particular, we demonstrate here for the first time that an identical global loss of visibility can evoke opposite percepts, depending on the nature of the underlying visual contrast reduction. Therefore, contrarily to what has been consistently reported in previous studies, a global contrast reduction can also lead to an overestimation of visual speed. This is notably the case when contrast is not reduced uniformly for all objects of the visual scene but varies according to their distance from the viewer. For instance, in fog, contrast reduction is more important for distant than for close objects. This generates a distance-dependent visibility gradient between the peripheral and central area of the visual field. Our results show that in this situation, perceived speed is not determined by the global level of contrast per se, but rather by the spatial distribution of contrast over the visual scene. More specifically, perceived speed is determined by the relative contrast between the central and peripheral areas of the visual field. When visibility is better in the peripheral than in the central visual field, as is the case in fog, speed is overestimated. Inverting the direction of the contrast gradient with anti-fog and thereby obscuring more the peripheral than the central region of the visual field, inverts the perceptual bias such that speed is now underestimated. This highlights the critical role of the visibility gradient in perceived speed, explaining why speed is unexpectedly overestimated in fog despite a global reduction of visibility. Importantly, our results also evidence the direct relationship between perceived and produced speed. Specifically, speed overestimation systematically prompted drivers to drive slower, whereas speed underestimation led to faster driving paces. This demonstrates that driving speed is strongly affected by perceived visual speed.

In 'real-life' driving, the roadsides usually include various objects and landmarks as trees, buildings, traffic signs, or pedestrians. Such objects next to the road can have a cognitive influence on driving speed because they constitute potential obstacles that increase the risk of collision (e.g., tree or building if one drives out of the road and cars or pedestrians suddenly crossing the road). However, we were interested in the perceptual and not in the cognitive effects of contrast reduction on driving speed. Therefore, the roadsides of our driving scenario consisted of a grass texture clear of any object and landmark. This prevented cognitive factors as those mentioned above to bias our results, but importantly, it also prevented subjects of using these landmarks to estimate moving speed. Indeed, adding landmarks or other 'higher-level' cues would have given the subjects the possibility to use some cognitive strategies to estimate speed. For instance, subjects could have used relative size information of landmarks to infer distances and counted passing them to assess speed. Undoubtedly, this would have biased our results. Yet, one could argue that at a 'pure' perceptual level, objects next to the road can also contribute to increase peripheral visual flow, thereby altering perceived speed. This would be mostly true with a driving environment that provides only little visual flow information (e.g., very smooth road pavement and roadsides). However, the scene we used consisted of a rough asphalt pavement and roughly textured grass roadsides (see *Figure 2*). These rough textures provided participants with rich visual flow information when driving, as attested by our results. Specifically, in the 'clear visibility' condition, participants estimated visual speed with great accuracy. In addition, the variability of speed estimates was small (i.e., low JNDs) in all conditions, and even more with clear visibility. Those elements indicate that our driving scenario provided robust visual flow information allowing for reliable speed estimates.

Contrast-dependent modulation of neural activity has been reported at early stages of visual processing, as the retina (*Shapley and Victor, 1978*), the lateral geniculate nucleus (*Solomon et al., 2002*), and the primary visual cortex (*Levitt and Lund, 1997*; *Polat et al., 1998*; *Sceniak et al., 1999*). It has also been reported in the middle temporal area (MT or V5), an area playing an important role in motion processing (*Pack et al., 2005*; *Bartels et al., 2008*) and speed detection (*Maunsell and Van Essen, 1983b*), notably via feedforward projections to the medial superior temporal area (*Maunsell and van Essen, 1983a*; *Ungerleider and Desimone, 1986*). Yet, linking these neurophysiological findings with the effect of contrast on perceived speed in humans is not straightforward. For instance, the observation that MT neurons tuned to high speeds are strongly activated by slow stimuli at low contrast (*Pack et al., 2005*) would predict speed overestimation at low contrast. This is the opposite of what has been usually observed with uniform contrast reduction, including in the current study. As stated by *Pack et al. (2005)*, this inconsistency might be resolved by assuming a bias towards slow speeds when the total MT population activity is low. Such a bias towards slow speed is precisely what

is proposed by Bayesian models of speed perception (*Weiss et al., 2002*; *Stocker and Simoncelli, 2006*). Specifically, these models rely on the assumption that speed 'measurements' are intrinsically noisy, and that based on our everyday experience, slower motions are more likely to occur than faster ones—resulting in an elevated prior for slow speeds. Whereas Bayesian models of speed perception can accurately predict speed underestimation at low contrast (*Stocker and Simoncelli, 2006*), speed overestimation observed at higher speeds (*Thompson et al., 2006*) and at low luminance (*Hammett et al., 2007*) are more difficult to account for. Alternative models have proposed that speed is encoded as 'the ratio of the responses of physiologically plausible temporal filters' (*Hammett et al., 2000*; *Hammett et al., 2005*). These 'ratio models' account for both under- and overestimation of speed at low contrast (*Thompson et al., 2006*). However, none of the abovementioned models addressed situations in which the amount of contrast reduction differed for different areas of the moving visual scene, as is the case when driving in fog. In that respect, these models do not explain the differences in speed perception observed here with uniform and distance-dependent contrast reduction. They do not account for opposite biases evoked with the same global contrast reduction at the same actual speed.

Poor visibility conditions affect millions of drivers around the world. Thousands of them die each year in a car accident. Excessive speed constitutes a major causal factor for these car accidents. We show here for the first time how fog biases speed perception, and we reveal the perceptual mechanisms underlying this bias, providing important insights into the human visual system. In particular, we show that contrarily to what was previously believed, speed is overestimated in fog because visibility is poorer in the central than in the peripheral area of the visual field. We also show that the behavioural consequence of this speed overestimation is a natural tendency to drive at a slower pace. Therefore, drivers should probably listen to their visual system when it prompts them to decelerate.

## Materials and methods

### Subjects

Thirty-two experienced drivers (23 males and 9 females; aged 21–35 years, mean = 25.3 years) participated voluntarily in the study (12 in experiment 1, 10 in experiment 2, and 10 in experiments 3 and 4). All had normal or corrected-to-normal vision. They were paid, naive as to the purpose of the research, and gave their informed consent before taking part in the experiment. The study was performed in accordance with the ethical standards laid down in the 1964 Declaration of Helsinki, in line with Max Planck Society policy and in compliance with all relevant German legislation. The participants had the option to withdraw from the study at any time without penalty and without having to give a reason.

### Experimental setup

All experiments were performed in an immersive virtual environment. For all experiments, the participants were seated in a simplified vehicle mock-up equipped with steering wheel and pedals. The steering wheel haptic feedback was disabled so that no speed information could be inferred from the wheel feedback. Also, the steering wheel rotation amplitude was linearly mapped to the vehicle turning speed and applied on the centre of mass of the vehicle. This way, a small rotation of the steering wheel resulted in a slow rotation of the vehicle around its vertical axis, whereas a large steering wheel rotation led to a fast turning, independently of the actual longitudinal speed of the vehicle. The mock-up was located at the centre of a large semi-spherical screen equipped with a multi-projection system. The panoramic screen surrounded the observer to provide an image that embraces almost the entire human visual field. More specifically, a cylindrical screen with a curved extension onto the floor provided a projection surface of 230° (horizontally) × 125° (vertically). The resulting surface was entirely covered by four LCD projectors, with a resolution of 1400 × 1050 pixels each. Overlapping regions were blended by openWARP technology (Eyevis, Reutlingen, Germany). The geometry of the scene was adjusted for an eye height of 0.8 m at a distance of 3 m from the vertical screen. The virtual environment was created using 3DVIA Virtools 4.1 (Dassault Systemes, Vélizy-Villacoublay, France) behavioural engine, which was running distributed on a 5-PCs cluster, one for each of the projectors and a supervisor. The visual stimulus consisted of a virtual environment where a textured plane reproducing a straight single-lane road was scrolled at different speeds. The central surface of the plane was a rough asphalt pavement, whereas the sides were covered with grass-like textures. The sky consisted

of a homogenous grey texture with the same colour of the fog and the plane used for the uniform contrast reduction (RGB = [128, 128, 128]).

## Contrast reduction

Visual contrast was reduced by blending the fog colour (RGB = [128, 128, 128]) into the virtual scene, according to the alpha blending model $C_r = C_o\,α + C_f\,(1 − α)$, where, for each pixel of the projected image, $C_r$ is the resulting colour, $C_o$ is the original colour, $C_f$ is the colour of the fog, and α is the blending factor. For the distance-dependent contrast reduction, the blending factor was determined by $e^{−f·d}$, where $f$ is the density of the fog and $d$ is the distance from the observer to the depicted object. The colour of each pixel was converted into the corresponding brightness (the arithmetic mean of the red, green, and blue colour coordinates in the RGB colour space), and the scene luminance distribution was computed based on the empirically determined function between luminance and brightness. The function was determined in two phases: (i) a plane model with uniform colour (brightness) was displayed on the screen facing the observer, and luminance was measured on its surface with a Minolta LS-100 photometer (Konica Minolta, Tokyo, Japan) for several brightness levels; (ii) the readings of the photometer were plotted against the corresponding brightness and fitted ($R^2 = 0.999$) by a quadratic function that was then used to compute the luminance distribution of the scene for each visibility condition. The contrast of the scene was then computed as the RMS of the luminance (in cd/m$^2$) of the pixels in the virtual scene (similarly to what *Snowden et al., 1998* did in their work). We set the fog density value to 0.1 for the medium visibility condition and 0.3 for the poor condition, in a range from 0 (full visibility) to 1 (no visibility). These values correspond to a meteorological visibility range (MVR) of 30 and 10 m, respectively. The MVR indicates the distance at which a white object appears with a 5% contrast (*Kovalev and Eichinger, 2004*).

The distance-independent (uniform) contrast reduction was implemented by inserting a transparent virtual plane in front of the scene. The plane brightness was composited with the brightness of the background image, according to the standard alpha blending model described previously. The opacity of the plane was adjusted to 0.28 and 0.52 in order to match the contrast of the moderate and severe fog conditions, respectively.

The global attenuation of visibility was identical for both types of contrast reduction (RMS = 0.46 for clear visibility, 0.31 for moderate contrast reduction, and 0.19 for severe contrast reduction). Anti-fog (experiments 3 and 4) was obtained using a vertex shader technique (*Engel, 2005*). The blending factor was set to $1 − e^{−af·d}$, where $af$ is the density of anti-fog, which was adjusted to match the overall scene contrast of the fog condition (RMS = 0.24).

## Design and data analysis

For the two-interval forced-choice (2IFC) procedure used in experiments 1 and 3, 80 trials per condition were performed, for a total of 1200 randomly interleaved trials in experiment 1 (two sessions of six blocks each, total duration of 2.5 hr) and 240 in experiment 3 (two blocks, total duration of 30 min).

For experiments 2 and 4, each subject performed five trials per condition, and mean driving speed was computed for each subject and condition. Experiment 2 consisted of 25 trials, performed in five consecutive blocks, and lasted 2 hr in total. Experiment 4 lasted 1.5 hr, consisting of three blocks of five trials each.

For all experiments, the order of presentation of the trials was fully randomized and different for all subjects. During the 5-min breaks between two consecutive blocks and the 15-min break between two sessions, the lights of the experimental room were switched on and subjects could walk and relax.

Mean PSE and JND values in experiment 1 and mean driving speed values in experiment 2 were analysed using a 5 × 3 (contrast reduction [clear visibility, moderate fog, severe fog, moderate uniform reduction, and severe uniform reduction], target speed [40, 60, and 90 km/hr]) repeated-measures (within subjects design) analysis of variance (ANOVA). Mean PSE and JND values in experiment 3 and mean driving speed values in experiment 4 were analysed using a 3 (contrast reduction [clear visibility, fog, and anti-fog]) repeated-measures ANOVA. The reported values are Huynh–Feldt corrected, and post hoc tests using the Holm adjustment method for multiple comparisons ($p < 0.05$) were performed when necessary.

## Acknowledgements

The authors thank Roland Fleming, Peter Banton, Muriel Lobier, and two anonymous reviewers for their comments on an earlier version of the paper and Joachim Tesch and Cora Kürner for technical support.

# Additional information

## Funding

| Funder | Grant reference number | Author |
| --- | --- | --- |
| Max Planck Society | | Paolo Pretto, Jean-Pierre Bresciani |
| World Class University program funded by the Ministry of Education, Science and Technology through the National Research Foundation of Korea | R31-10008 | Heinrich H Bülthoff |

The funders had no role in study design, data collection and interpretation, or the decision to submit the work for publication.

## Author contributions

PP, Conception and design, Acquisition of data, Analysis and interpretation of data, Drafting or revising the article; J-PB, Conception and design, Acquisition of data, Analysis and interpretation of data, Drafting or revising the article; GR, Analysis and interpretation of data, Drafting or revising the article; HHB, Analysis and interpretation of data, Drafting or revising the article.

## Ethics

Human subjects: The work was conducted in line with Max Planck Society policy and in compliance with all relevant German legislation. The study did not require specific approval by an ethics committee. However, the ethical aspects of the study were carefully discussed with local researchers not involved in the experiments and prior to the study. The research did not involve serious risk for participants and did not make use of invasive techniques. The participation in the study was voluntary and participants were recruited using the Max-Planck Subjects Database. Before starting the experiments, an informed consent form was signed by participants, who were also informed that withdrawal from the study was possible at any time, without any penalty, and without having to give a reason. Participants data protection complied with the German Federal Data Protection Act.

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
