## [Author Response]

*1) The fact that each of the figures depicted a different amount of scene detail raised questions about the actual amount of detail in the stimuli and the ecological validity of the scene*.

*a) Figure 1 shows a textured road below a dark sky. Figure 2 shows a road bright clouded sky and additional details like trees and fences. Supplemental Figure 1 shows the road and bright clouded sky but without additional landmarks. The revised manuscript should unequivocally describe the amount of detail in the actual scene employed in the experiments and ideally should make each of the figures consistent with this (or at minimum should include a caveat in the figure caption). The consensus was to discourage the use of supplemental figures unless absolutely necessary. A figure showing the actual stimuli would be more informative in the main text than Figure 2. The extant Figure 2 could be merged with Supplementary Figure 1 or deleted*.

The scene we used provided rich visual flow information but was clear of any landmarks to prevent subjects using cognitive strategies to estimate moving speed. Following the suggestion of the reviewer, the following modifications have been made in the revised version of the manuscript:Figure 1 and 2 show the actual stimuli used for the experimentThe visual stimulus is described in more details in the Materials and Methods section (Experimental setup subsection)Two pictures of the actual set-up and stimuli are provided as they give a more realistic impression of the road texture and of the large field of view in the simulator.

*b) If details like the sky luminance actually varied, the manuscript should describe whether the overall luminance had an impact on speed perception and driving speed*.

The different skies depicted in the figures are misleading. Indeed, the sky was the same in all experiments and consisted of a homogenous grey texture with the same color of the fog and the plane used for the uniform contrast reduction (RGB = [128, 128, 128]). The overall luminance of the scenes was measured via photometer and the fog intensity was adjusted in order to match the corresponding uniform contrast condition, as described in the Materials and Methods section, Contrast reduction subsection. Figure 1 and Figure 2 have been changed and show now the actual stimuli used in the experiments.

*Given that any of the three scenes had impoverished detail compared to real life driving (e.g., there are few landmarks, traffic signs, lane markings or potential obstacles likes cars, bicycles or pedestrians, curves, hills, etc.), the authors should discuss the extent to which the driver's ability to estimate self-motion speed could be affected by the choice of stimuli compared to a real driving scenario*.

Drivers usually focus on the road ahead and its near surroundings (at least they should!). This is even truer when driving in fog ([23], *Nature*). And some studies have shown that road texture is of primary importance in visual speed perception (Denton, 1973; Agent, 1980). Accordingly, we used visual stimuli (i.e., rough road pavement and roughly-textured roadsides) that provided rich visual flow information concerning moving speed. Our results confirm that these stimuli indeed provided good moving speed information and allowed for reliable speed estimation as:Subjects estimated visual speed with great accuracy in the ‘Clear’ condition (control condition)The variability of speed estimates was small in all conditions.

Adding landmarks or other ‘higher-level’ cues would have given the subjects the possibility to use some cognitive strategies to estimate speed. For instance, subjects could have used relative size information to infer distances and counted to assess speed, which would have biased our results.

These aspects are discussed in the discussion of the revised version of the manuscript.

**References:**

Agent, K.R. 1980. Transverse pavement markings for speed control and accident reduction. *Transportation Research Record*
**773**, 11–14

Denton, G.G. 1971. The influence of visual pattern on perceived speed. *Transport and Road Research Laboratory Report, LR409*

*2) The presentation of the statistics should make it clearer how the data were analyzed and which differences were significant (for comparisons between conditions and clear viewing and for comparisons amongst conditions). Error bars are presented but are unspecified and cannot be used to interpret significance of condition differences for repeated measures designs*.

In the revised manuscript, the captions of Figure 3 and 4 specify that error bars represent the standard error of the mean for each condition (i.e., between subjects variability).

*The specifics of the ANOVAs are missing (factorial or not; outcome of main effects and interactions)*.

This information is provided in the Materials and Methods section (Design and data analysis subsection) of the revised version of the manuscript.

*3) The revised manuscript should describe the extent to which the haptic feedback from the steering wheel gave the driver information about vehicle speed during the driving condition. Surely a slight rotation of the steering wheel at 20 km/h will have a different effect on the simulated car position in the scene than the same rotation at 120 km/h. Could the drivers be using this information to help formulate their speed judgements and/or to adjust their driving speed*?

The virtual vehicle was designed to avoid any possible speed cue deriving from the vehicle dynamic behavior. The steering wheel haptic feedback was disabled so that no speed information could be inferred from the wheel feedback. Also, the steering wheel rotation amplitude was linearly mapped to the vehicle turning speed and applied on the center of mass of the vehicle. This way, a small rotation of the steering wheel resulted in a slow rotation of the vehicle around its vertical axis, while a large steering wheel rotation led to a fast turning, independently of the actual longitudinal speed of the vehicle.

This description has been added to the revised manuscript.

*4) The experimental sessions lasted for long periods. The revised manuscript should discuss the extent to which contrast adaptation could affect performance over the trials. Was there any evidence for sequence effects over trials (e.g. from the beginning to the end of a session) that could be related to prolonged exposure to different contrast levels*?

Contrast adaptation is very unlikely to have biased our results because:- There were 3 different levels of contrast (clear, moderate, severe) in experiment 1 and 2 and 2 different levels (clear, reduction) in experiment 3 and 4.- When contrast was reduced, the type of reduction was different (uniform vs fog and fog vs anti-fog).- The order of presentation of the conditions was fully randomized, and was different for every subject.

In addition, each experiment included regular breaks during which the lights of the room were turned on for 5 minutes, followed by 3 minutes dark adaptation before starting a new session.

Some of these details have been added to the Materials and Methods section, Design and data analysis subsection of the manuscript.

*Why did the trials in Exp. 4 last so long (3 x 5 trials over 1.5 hrs)? Please explain*.

As for experiment 2, the total duration of the experiment considers the time from when the participants enter the experimental room to when the experiment is finished. This includes the time to accommodate the participants, provide the instructions, run the training (Figure 1B, left panel), and the time of the actual experiment (Figure 1B, right panel). During the experiments a 5 minute break was also provided between each session. The time reported in the manuscript is the maximum time it took to complete the experiment.